# Unveiling the Impact of Urbanization on Net Primary Productivity: Insights from the Yangtze River Delta Urban Agglomeration

Jing Gao [1], Min Liu [2,3] and Xiaoping Wang [4,*]

1 School of Teacher Education, Northwest Normal University, Lanzhou 730070, China; gaojing@nwnu.edu.cn
2 Shanghai Key Laboratory for Urban Ecological Processes and Eco-Restoration, School of Ecological and Environmental Sciences, East China Normal University, Shanghai 200241, China; mliu@re.ecnu.edu.cn
3 Technology Innovation Center for Land Spatial Eco-Restoration in Metropolitan Area, Ministry of Natural Resources, Shanghai 200241, China
4 State Key Laboratory of Soil Erosion and Dryland Farming on the Loess Plateau, College of Natural Resources and Environment, Northwest A&F University, Xianyang 712100, China
* Correspondence: wxp4911@nwafu.edu.cn; Tel.: +86-130-5752-5010

**Abstract:** Urbanization has significantly altered the carbon cycle of the terrestrial environment, particularly in relation to net primary productivity (NPP). Gaining a more comprehensive comprehension of how NPP is affected by urbanization is crucial for obtaining fresh perspectives on sustainable urban landscape design and decision making. While there is a significant body of research examining the geographical and temporal patterns of NPP supply capacity, there are only a few studies that have investigated the spatial relationships between NPP and urbanization, particularly at the grid scale. This research investigated the temporal and geographical features and patterns of NPP and their impact mechanisms. In order to estimate NPP and the level of urbanization in the Yangtze River Delta Urban Agglomeration (YRDUA), we used a combination of different models and datasets. To evaluate the geographical correlations and dependence between NPP and urbanization, we utilized local bivariate autocorrelation methods and spatial regression models to describe and visualize these relationships. The findings revealed that there was a consistent negative relationship between NPP and urbanization on a global scale from 1990 to 2020. However, when examining the local scale, the geographical correlations could be classified into four distinct categories: areas with both low NPP and low urbanization, areas with high NPP and high urbanization, areas with low NPP and high urbanization, and areas with high NPP and low urbanization. Our analysis showed that spatial regression models are more suitable for quantifying the spatial relationship between NPP and urbanization due to their ability to include the impacts of spatial Moran's I techniques. Due to the growing urbanization, the highest NPP value was recorded in 2005, followed by 2000, 2020, and 2010. Conversely, the smallest association was observed in 2015. Examining the geographical connection between NPP and urbanization offers theoretical and practical insights for urban planning that prioritizes human needs and promotes sustainable development. It also aids in the development of reasonable methods for organizing ecological functional systems.

**Keywords:** NPP; urbanization; spatial dependence; spillover effects; urban agglomeration

## 1. Introduction

Urban agglomerations have caused the conversion of natural ecosystems into ecosystems that are either dominated by humans or closely connected to human activities [1]. The process of urbanization is primarily driven by population concentration, economic growth, and urban expansion. These factors are recognized as the key drivers of changes in NPP within urban agglomerations [2]. Urbanization often leads to transformations in land

use and land cover (LULC), affecting not just metropolitan areas but also their surrounding regions. For instance, the substitution of vegetation areas and the implementation of urban greening may directly change the composition of local terrestrial ecosystems [3,4]. Furthermore, urbanization has significantly impacted the environment for plant growth, including factors such as temperature, soil texture, and atmospheric conditions. Net primary production (NPP) is a common consequence of urbanization and has always been a subject of significant study interest [5]. Vegetation NPP refers to the total amount of organic matter produced by photosynthesis, minus the organic matter consumed by respiration; it represents the total amount of organic matter accumulated by vegetation per unit area and per unit time [6]. The dynamic changes in NPP, as a key parameter of terrestrial ecological processes and an important indicator reflecting the regional ecological conditions, can reflect the impact of climate change and human activities on ecosystems [7]. Studying the spatial–temporal patterns and driving factors of vegetation NPP is of great significance for the protection of regional ecological environments and sustainable development [8]. Human activities have a significant role in ecological management, since actions such as irrigation, pruning, and tree cutting may have substantial impacts [9]. In addition, urbanization may significantly alter terrestrial ecosystems, particularly the carbon cycle systems within them, due to the aforementioned effects [10]. Over the last several decades, there has been a significant increase in urbanization worldwide, making it a crucial aspect of global transformation [11]. Gaining a deeper understanding of how urbanization affects NPP of terrestrial ecosystems is crucial in this specific context [12].

Ecosystem services refer to all the benefits that humans derive from ecosystems, which are categorized into four distinct groups: supply services (such as providing food and pure water), regulatory services (such as controlling floods and diseases), cultural services (such as entertainment and cultural benefits), and support services (such as maintaining nutrient cycling) [13,14]. The regulatory services act as a conduit that links the NPP of the environment with the welfare of humans. It mostly pertains to the functions of climate control, such as carbon fixation, oxygen release, and cooling impacts. The global NPP relies on regulatory services as a crucial component and essential connection, which contribute significantly to the overall global carbon equilibrium [15]. Hence, within the framework of global climate change, investigating alterations in vegetation NPP has immense importance in comprehending the interplay between variations in plant productivity and climate [16,17]. Historically, the study of NPP has mostly relied on quantitative methods, such as biometric assessments including sample surveys and field measurements [18]. Nevertheless, these conventional measures conducted in the field often require a significant amount of time and effort, making them challenging to implement on a large scale to estimate NPP. Models have been extensively used in recent decades to obtain more precise NPP estimates on broader temporal and spatial scales; these models include statistical [19], process-based [20], and light energy utilization [21] models. Researchers have used NPP simulation models to study the effects of urbanization and LULC changes on NPP. Imhoff et al. used the Carnegie Ames Stanford Approach (CASA) model to examine the consequences of urban land conversion in the United States. Their findings indicate that urbanization has significantly and detrimentally affected NPP [22]. Paz-Kagan et al. used NPP as a measure to evaluate the impact of land-use changes on the ecosystems in semi-arid regions of Israel [23]. In China, many scholars have used the CASA model to assess the temporal and geographical NPP patterns and the influence of urban growth on NPP [24–26].

The changes in NPP in terrestrial ecosystems are a clear indicator of the impact of both human activities and global climate change on vegetation. These changes have a significant effect on the global carbon cycle and climate change. The capacity of the earth to support life and the sustainable evolution of terrestrial ecosystems can be evaluated by using this indicator [27]. Zhao et al. utilized the Moderate Resolution Imaging Spectrometer MOD13A2 Enhanced Vegetation Index (EVI) product to quantify the changes in NPP and found that plant growth in most Chinese cities saw substantial improvements as a result of indirect factors [28]. This improvement offset approximately 40% of the losses resulting

from direct effects. Peng et al. used spatial regression to quantify the linear correlation between NPP changes and the three indicators of urbanization. They also identified the threshold at which NPP changes respond to these indicators [29]. Su et al. used the spatial lag model (SLM) to enhance the visualization of the non-stationary correlation between environmental services and urbanization [30]. While these studies attempted to examine the correlation between NPP and urbanization, several elements remain unexplored. There is a lack of consideration for the spatial relationship between NPP and urbanization, particularly at the regional level. Hence, other statistical methods must be used to address spatial autocorrelations. Furthermore, the previous research mostly concentrated on a single urban area, often using administrative districts to represent spatial entities. This level of study is insufficient to capture the spatial phenomena occurring at the meso or macro level, such as those occurring in towns, counties, and cities. This might restrict the practical feasibility of incorporating the NPP impact into comprehensive regional landscape design and the industrial arrangements of urban agglomerations.

The YRDUA is one of China's three main urban agglomerations and has the greatest economic growth rate and population density in the country. Over the last several decades, urbanization has caused significant changes in the land-cover conditions in the YRDUA, altering the structure and function of its terrestrial ecosystems. This process has significantly impacted the carbon budget of the area [31]. Hence, it is crucial to conduct more research on the impact of urbanization in the YRDUA on its NPP. The MOD17A3HGF V061 data products obtained from the data distribution system of the National Aeronautics and Space Administration (NASA) website offer NPP datasets with a resolution of 500 m. These datasets cover the period from 2000 to 2020 and fulfill the requisite criteria for both temporal duration and spatial precision.

The purpose of this study was to (1) use various models and multi-source data to quantify and map the degree of comprehensive urbanization, and analyze its spatial–temporal evolution pattern; (2) examine the relationship between urbanization and NPP using bivariate global and local Moran's I approaches; and (3) investigate the geographical relationship between urbanization and NPP, as well as other relevant parameters, using spatial regression models such as ordinary least squares (OLS) regression models and geographic weighted regression (GWR) models.

## 2. Materials and Methods

### 2.1. Study Area

Our study chose 16 prefecture-level cities as the research object, which are the core area of the YRDUA (Figure 1). The Lower Yangtze River, which borders both the East China Sea and Yellow Sea, is home to the central region of the YRDUA, which is located at 118° E–123° E, 28° N–33° N, and has an area of 167 thousand km$^2$, accounting for 1.74% of the total national land area. It is a segment of the alluvial plain near the point where the Yangtze River flows into the ocean, with an altitude of more than 10 m and low hills scattered between 200 and 300 m. The gross domestic product (GDP) of the 16 cities in the YRDUA's central area reached CNY 9.47 trillion in 2020 or 11.43% of the country's GDP. At this time, there were 119 million people living there, making up 9.68% of the entire population of the country. According to statistics data, the energy consumption of the 16 prefecture-level cities in the YRDUA core region surpassed 6869 Mt in 2020, constituting 15.92% of the overall energy usage in China.

### 2.2. Data Sources

We combined a variety of data sources that were diverse in nature, including both geographical and statistical feature data, into our analysis. More precisely, the datasets used were (1) yearly net primary production (NPP) data from the MOD17A3HGF V061 products with a spatial resolution of 500 m, obtained from the National Aeronautics and Space Administration (NASA) (https://lpdaac.usgs.gov/, accessed on 10 February 2021) [32]; (2) Land-use/land-cover (LULC) data from 1990 to 2020, compiled using a conventional

interpretation method that analyzed Landsat Thematic Mapper (TM) and Landsat 8 OLI remote-sensing imagery with a 30 m resolution. The LULC data were derived from Landsat scenes covering path/row designations 118–120 and 37–40, and they achieved a classification accuracy of over 95% based on confusion matrix and Kappa coefficient testing [33,34]; (3) Gridded datasets of GDP and population at a 1 km spatial resolution, obtained from the Resource and Environment Science and Data Center (RESDC) of the Chinese Academy of Sciences (https://www.resdc.cn/, accessed on 12 June 2021) [35]; (4) Meteorological datasets including annual air temperature and annual rainfall from 1990 to 2020, interpolated to a 1 km grid from observations at 83 weather stations situated across and around the Yangtze River Delta Urban Agglomeration (YRDUA) region. These meteorological data were acquired from the China Meteorological Data Service Center (http://data.cma.cn, accessed on 18 August 2021) [36]; (5) Digital elevation model (DEM) data at 90 m resolution, resampled to 1 km resolution, obtained from the Geospatial Data Cloud platform of the Computer Network Information Center, Chinese Academy of Sciences (https://www.gscloud.cn/, accessed on 18 April 2021) [37].

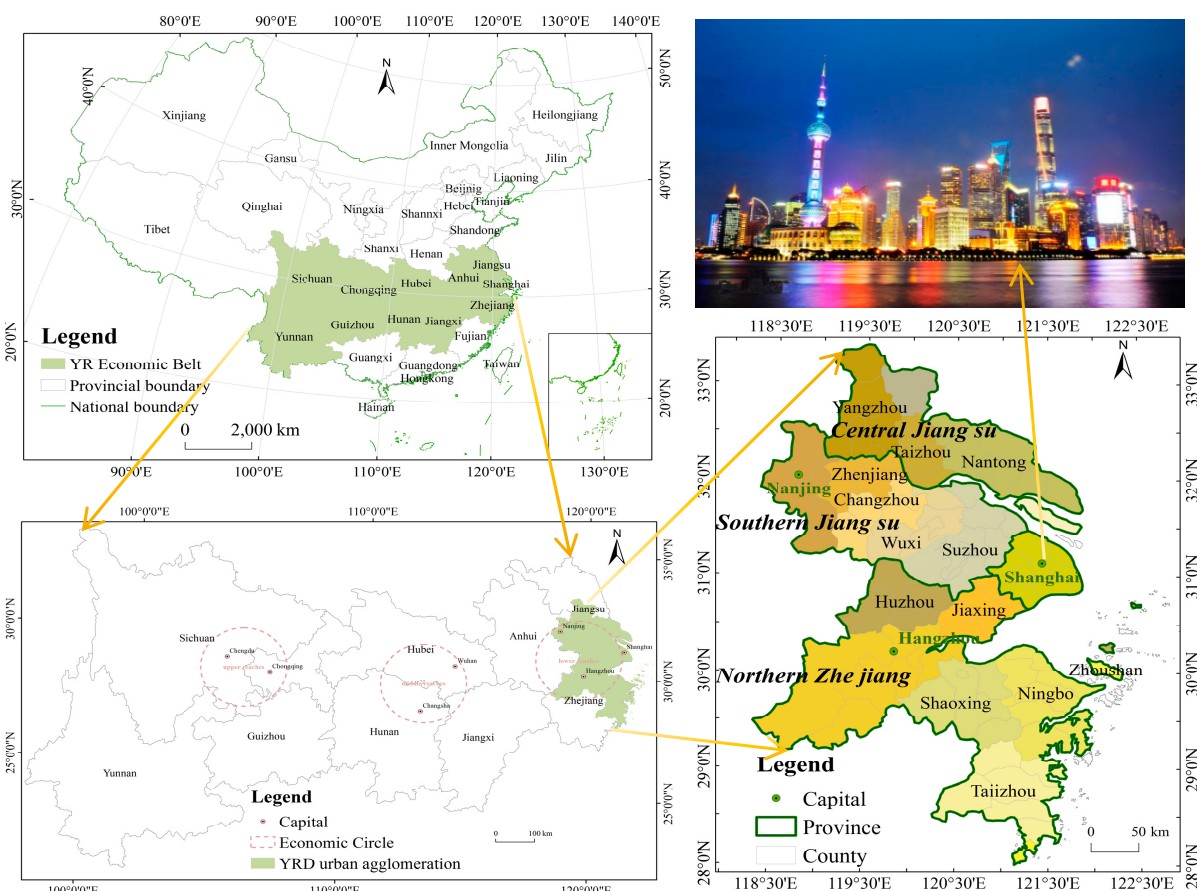

**Figure 1.** Map of the study area. **Top left** panel: the national boundary of China (green area is the location of the YR Economic Belt in China); **bottom left** panel: the administrative boundary of the YR Economic Belt (green area is the YRDUA in the YR Economic Belt); **bottom right** panel: the administrative boundary of the YRDUA (administrative districts); and **top right** panel: a representative city (Shanghai) in the YRDUA.

## 2.3. Data Analyses and Methods

The process of determining the NPP response to urbanization mainly included the following three steps: (1) quantitative characterization of the degree of comprehensive urbanization; (2) spatial correlation analysis between the changes in NPP and urbanization; and (3) spatial regression analysis of NPP and urbanization. A flow chart of the procedure is shown in Figure 2.

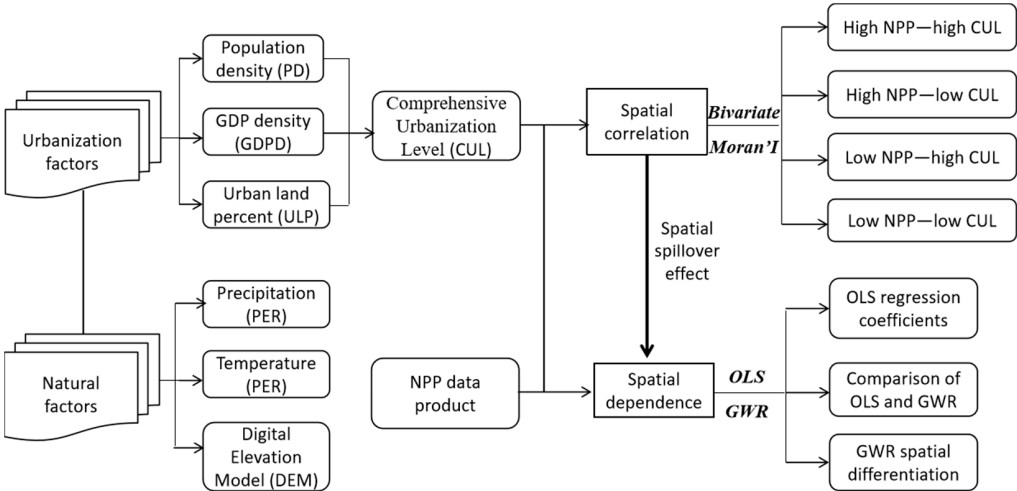

**Figure 2.** The process for determining the spatial relationship between NPP and urbanization.

2.3.1. Urbanization Assessment

The urbanization process can be generally characterized by the growth of the population, increase in the total economy, continuous improvement in the quality of life, and rapid growth of urban construction areas. In view of the fact that social urbanization data are not easy to collect and the indicators are relatively complex, we did not consider these data here; thus, the degree of urbanization was measured through the three other aspects. More precisely, population density (PD) was used as a metric to quantify the degree of urbanization in terms of population, gross domestic product density (GDPD) was selected to reflect the economic development level, and urban land percentage (ULP) was utilized to gauge the extent of urbanization in terms of land usage. Due of the high similarity in the geographical patterns of PD, GDPD, and ULP, these three variables were combined into a single indicator known as comprehensive urbanization level (CUL). The various indices were subjected to range standardization in order to convert their values into a uniform range of 0 to 1. These standardized values were then averaged to obtain the CUL value. The range standardization method (Equation (1)) and CUL calculation (Equation (2)) are as follows:

$$U_{i,j}^{'} = \frac{U_{i,j} - U_{i,min}}{U_{i,max} - U_{i,min}} \tag{1}$$

$$CUL_j = \left(PD_j + GDPD_j + ULP_j\right)/3 \tag{2}$$

where $U_{i,j}^{'}$ represents the normalized value of $U_{i,j}$; $U_{i,j}$ is the *i*-th urbanization indicator (PD, GDPD, or ULP) in the *j*-th raster, relative to the original value; $U_{i,max}$ and $U_{i,min}$ represent the highest and lowest values, respectively, of the *i*-th urbanization indicator over all grids; $CUL_j$ represents the urbanization level of the i-th grid; and $PD_j$, $GDPD_j$, and $ULP_j$ represent the population density, GDP density, and urban land proportion, respectively, of the *j*-th grid after standardization. The rationale for using Equation (2) to calculate the CUL is that it provides a balanced and holistic measure of urbanization by equally weighting the three key aspects [38].

2.3.2. Spatial Correlation Measure

The bivariate Moran's I statistic was utilized to identify any geographical clustering or discontinuous link between the comprehensive urbanization level (CUL) and vegetation net primary production (NPP). Global and local bivariate Moran's I are two strategies that can be used for this purpose. The formulae that were used are as follows (Equations (3a) and (3b)):

$$I_{cu} = \frac{N\sum_i^N \sum_{j \neq i}^N W_{ij} z_i^c z_j^u}{(N-1)\sum_i^N \sum_{j \neq i}^N W_{ij}},\tag{3a}$$

$$I'_{cu} = z^c \sum_{j=i}^N W_{ij} z_j^u,\tag{3b}$$

Here, $I_{cu}$ and $I'_{cu}$ are the global and local bivariate Moran's I of NPP and CUL, respectively, and $N$ represents the aggregate number of spatial grid cells. In the model, the parameters obtained are $W_{ij}$, which represents an $N \times N$ weighted matrix that was used to detect the correlation between the $i$-th and $j$-th grids. The spatial unit is a $4 \times 4$ matrix generated based on the first-order neighborhood in the weight adjacent to the queen [39]. The input data were $z_i^c$ and $z_j^u$; $z_i^c$ represents the $i$-th standardized NPP grid value obtained by using Equation (1), and $z_j^u$ represents the $j$-th standardized CUL unit value calculated using Equation (1) [40,41]. The output result is $I_{cu}/I'_{cu}$, where the range of values for $I_{cu}/I'_{cu}$ is $-1$ to 1. The computed $p$-value for the regional connection between the NPP impact and CUL was below 0.05, indicating statistical significance [42]. NPP and CUL were readjusted to a $1 \times 1$ km raster map using the mean value approach in ArcGIS 10.5. Next, the NPP and CUL data of all grids were entered into GeoDa 1.12 (https://geodacenter.github.io/, accessed on 16 August 2021) for execution, and spatial correlation analyses were conducted [43].

Bivariate spatial autocorrelation can determine whether two variables are spatially correlated and evaluate the strength and direction of the correlation. It can help us in exploring the laws of geographical phenomena and spatial distributions, providing a scientific basis for decision making [44]. However, bivariate spatial autocorrelation analysis also has some limitations, since it is sensitive to data distribution biases, spatial scale effects, spatial connections, and causal relationships. Therefore, we should take these limitations into consideration when interpreting the results [45,46].

2.3.3. Spatial Regression Test

1. Analysis of global spatial regression

Ordinary least squares (OLS) can generate predictions when performing global linear regression, or model a dependent variable and a set of explanatory variables to detect the influence relationship. Anselin provides the general form of the spatial regression equation for raster data, taking into account the spatial correlation between independent variables and dependent variables [47] (Equations (4) and (5)):

$$Y = \rho W_1 Y + X\beta + \varepsilon,\tag{4}$$

$$\varepsilon = \lambda W_2 + \mu, \mu \sim N(0, \Omega), \Omega_{ii} = h_i(za),\tag{5}$$

where $\rho$ represents the coefficient of the geographical lag variable $W_1 Y$; $\beta$ represents the $k \times 1$ parameter vector associated with the independent variable X; $\varepsilon$ is the vector representing the random error term; the weight matrix $W_1$ represents the geographical pattern of the variable; the order weight matrix $W_2$ represents an $n \times n$ matrix; the normal distribution is denoted by N; the exogenous variable is represented by z, while $\Omega$ denotes the variance matrix, its diagonal elements are $\Omega_{ii}$, $h_i$ is the functional relationship, and the constant term is represented by a; and the spatial autonomy is denoted by $\lambda$. The coefficients of the regression structure $W_2$ should generally be $0 \leq \rho < 1$, $0 \leq \lambda < 1$, and $\mu$ is a random error vector of a normal distribution. The regression equation of the whole grid data space is subject to 3 parameters: $\rho$, $\lambda$, and $a$.

2. Analysis of local spatial regression

Spatial regression technology was used to study the spatial dependence of the effect of urbanization on NPP (that is, how NPP changes in response to the process of urbanization). Geographically weighted regression (GWR) is a type of regression that adds regional

ordinary least squares (OLS) to improve the model [48]; the expression of the model is as follows (Equation (6)):

$$\text{ZFJG}_i = \beta_o(u_i, v_i) + \sum_{i=1}^{T} x_u \beta_i(u_i, v_i) + \varepsilon_i, \tag{6}$$

where $\beta_o(u_i, v_i)$ is a constant term; $\beta_i(u_i, v_i)$ is the characteristic elastic coefficient of the i-th sample point. The elastic coefficient of every point $(u_i, v_i)$ in the sample region is determined using a weighted least square multiplication method; the calculation formula is as follows (Equation (7)):

$$\hat{\beta}(u_i, v_i) = \left(\hat{\beta}_\rho(u_i, v_i), \hat{\beta}_\tau(u_i, v_i)\cdots, \hat{\beta}_\gamma(u_i, v_i)\right)^T = (X^\gamma W(u_i, v_i)X)^{-1}X^\gamma W(u_i, v_i)\text{ZFJG}_i \tag{7}$$

where X represents the matrix of independent variables and $W(u_i, v_i)$ represents the spatial weight matrix. The spatial weight matrix is constructed using a monotonically decreasing function that calculates the geographical distance between the location to be estimated and the surrounding observation sites. Different function forms can be used. Our study used the Gauss kernel function; its expression is as follows (Equations (8) and (9)):

$$W(u_i, v_i) = \text{diag}\left(\text{K}(d_{io}/h), \text{K}(d_{i\tau}/h)\cdots, \text{K}(d_{i\gamma}/h)\right), \tag{8}$$

$$\text{K(t)}\frac{1}{\sqrt{2\pi}}\exp\left(-\frac{1}{2}t^2\right), \tag{9}$$

where $d_i$ is the Euclidean distance between each sample point and $h$ is the optimal bandwidth, which can be determined using the cross-determination method to minimize h (Equation (10)).

$$\text{CV(h)} = \frac{1}{n}\sum_{i=1}^{n}\left(\text{ZFJG}_i - \text{Z}\hat{\text{F}}\text{JG}_{(-i)}(h)\right)^2, \tag{10}$$

where $\text{Z}\hat{\text{F}}\text{JG}_{(-i)}(h)$ is the simulated predicted value of the NPP at point $i$ obtained by simulation after the $i$-th observation value is discarded under $h$, and $\text{ZFJG}_i$ is the actual observed value of the NPP at point $i$.

## 3. Results

### 3.1. Spatial CUL Patterns in the YRDUA

According to the analysis of Figures 3 and 4 and Supplementary Materials, from 1990 to 2000, the comprehensive urbanization level of the YRDUA showed a predominant pattern of circular expansion around the major cities. During the two five-year periods from 1990 to 1995 and 1995 to 2000, Shanghai, as the main city in the urban agglomeration, saw significant urbanization development. The comprehensive urbanization level increased from 67.37 to 105.75, with growth rates of 6.05% and 12.22%, respectively. The comprehensive urbanization level developed slowly in the YRDUA.

After 2000, the YRDUA achieved axial expansion, i.e., expansion of the circle around core cities. The urbanization level of the growth poles and key cities on the Nanjing–Hangzhou Expressway increased significantly. The growth pole city Hangzhou had an added value of urbanization of 130.86, with a growth rate of 67.11%. The comprehensive urbanization growth values of key cities such as Ningbo, Huzhou and Shaoxing were 66.21, 21.55, and 51.73, and the growth rates were 120.32%, 41.66%, and 87.43%, respectively. The process of urbanization developed rapidly.

After 2010, as the development of urban agglomerations continued to increase, cities within urban agglomerations gradually obtained their own independent development space. The linear connection mode was gradually replaced by the expand-around mode, and the evolution of urban agglomerations was transformed into a network mode. Due to the changes in macro policy and concept aspects, such as adjusting industry structures and promoting upgrades, Shanghai and Hangzhou have maintained a steady growth rate due to the development of their emerging high-tech industries [49]. The added values of

urbanization were 104.78 and 73.93, respectively, with growth rates of 9.35% and 17.39%; thus, the urbanization process has grown steadily.

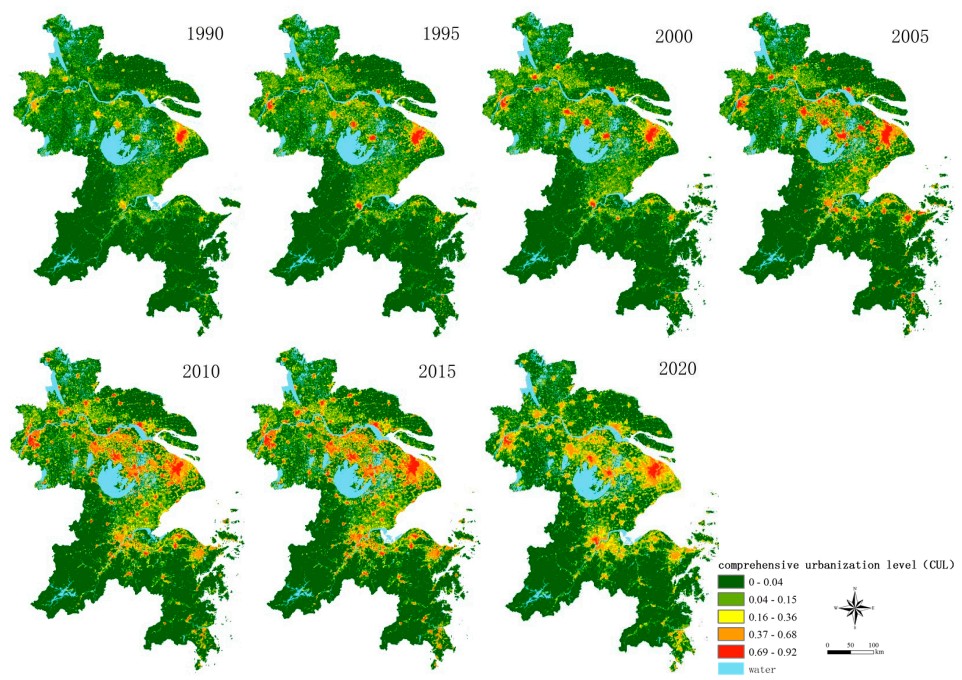

**Figure 3.** Spatial pattern of comprehensive urbanization level (CUL) in the YRDUA from 1990 to 2020.

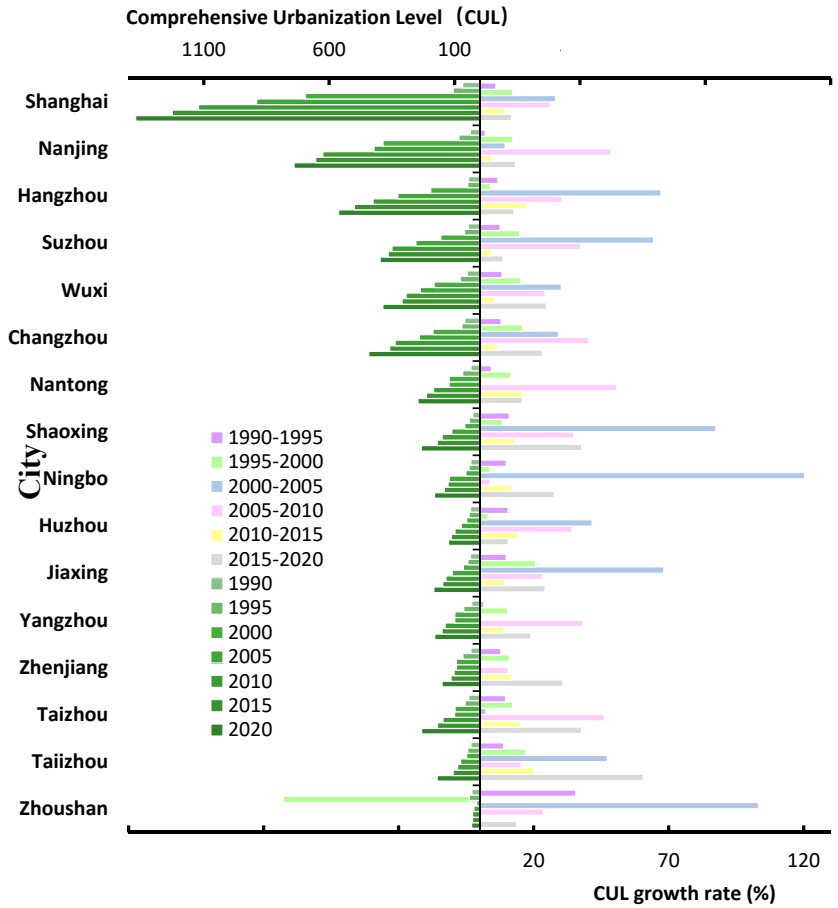

**Figure 4.** Comparison of comprehensive urbanization level (CUL) across the 16 cities in the YRDUA during 1990–2020.

### 3.2. Geographical Links between Urbanization and NPP

Moran's I analysis revealed a notable negative geographical association between the NPP and urbanization, irrespective of the year (Figure 5). It is known that the growth of built-up land led to a decline in NPP at the global level. Nevertheless, the extent of the negative association varies depending on the various phases of urban agglomeration growth. Here, the association between NPP and urbanization was the strongest in 2010 (Moran's I: −0.2492), followed by 2020 (Moran's I: −0.1937), 2015 (Moran's I: −0.1841), 1990 (Moran's I: −0.1685), 2000 (Moran's I: −0.1685), and 2005 (Moran's I: −0.1470). The weakest correlation was in 1995 (Moran's I: −0.1234). The results of the global spatial autocorrelation analysis, to some extent, showed a spatial correlation between NPP and CUL, and overall, the negative correlation increased over time.

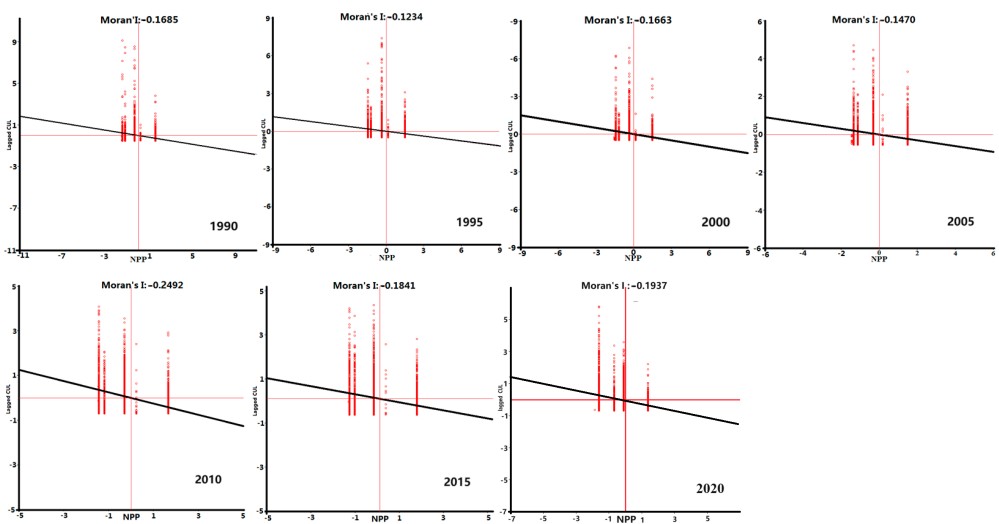

**Figure 5.** Spatial–temporal correlations between NPP and CUL (global bivariate Moran's I autocorrelation).

Local bivariate spatial autocorrelation is a statistical method used to analyze the spatial correlation between two variables at the local level; it introduces the spatial concept into the autocorrelation analysis, allowing us to determine whether two variables are spatially correlated and the strength and direction of the correlation, which are usually represented by the local indicators of spatial association (LISA) [50]. The LISA diagram shows the four possible geographical correlations that exist between urbanization and NPP (Figure 6): the high–high (HH) type represents the clustering of high NPP and high CUL values; the low–low (LL) type represents the clustering of low NPP and low CUL values; the low–high (LH) clustering represents the clustering of low NPP and high CUL values; and the high–low (HL) clustering represents the clustering of high NPP and low CUL values [51]. Using a seven-year sample, we saw distinct similarities in the way NPP and urbanization were clustered in different regions. The places with the highest elevation are mostly located in the central regions of the urban land of the YRDUA. With the expansion of urban land, HH areas also increased. The LH regions were mostly dispersed over the whole HH region and concentrated around the HH areas. The low–high regions were mostly concentrated in the northern region of the urban agglomeration, whereas low–high regions were absent from the southern region. The HL area occupied a large area in the south, concentrated in the mountains in southwest Hangzhou and the lush vegetation areas in the southern mountainous areas of Taizhou, Ningbo, and Shaoxing, which had relatively low levels of urbanization due to being restricted by natural conditions such as topography and landforms. The LL area did not show any changes in its spatial pattern over time. In 2005, 2015, and 2020, the LL areas appeared in the coastal areas of Hangzhou, Shaoxing, and Ningbo, while in the other four years, the LL areas did not show any obvious spatial characteristics or LL areas did not appear.

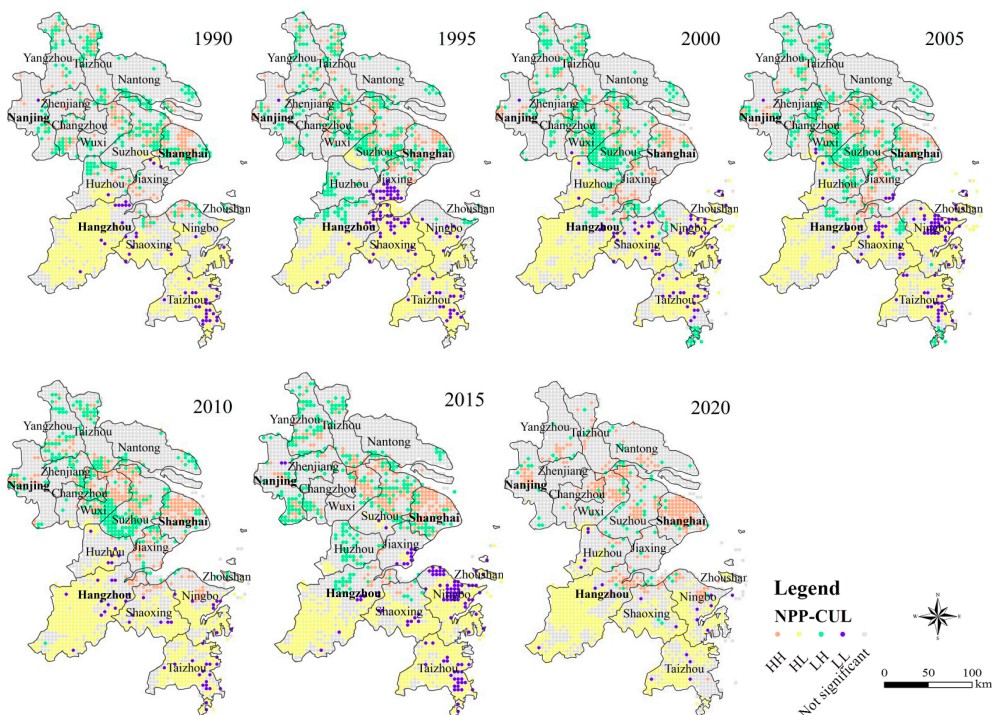

**Figure 6.** Spatial–temporal correlations between NPP and CUL (LISA diagram).

### 3.3. Spatial NPP Pattern Dependence on Urbanization

Changes in the ecosystem are fundamental impacts of climate change; therefore, climatic factors (including temperature, precipitation, and digital elevation models (DEM)) were analyzed in order to reveal the driving factors affecting the regional NPP. The results simulated by the OLS model showed that the regression coefficients of precipitation in all years were positive, indicating that precipitation and the NPP were positively correlated (Table 1). Except for the regression coefficients of PD in 2000 and ULP in 2005 that showed positive correlations, all factors from 1990 to 2020 were negatively correlated with NPP. From 1990 to 2010, the absolute value of the regression coefficient of the CUL was always greater than that of the other factors (PD: −0.48; GDPD: −0.61; ULP: −0.53; TEM: −0.86; PRE and DEM: −0.56). GDPD ranked second in 1990 and 2000, with coefficients of −0.55 and −0.48, and PD in 1995 and 2005 ranked second with coefficients of 0.27 and −0.61, respectively. As the pace of urbanization stabilized, the population and economic growth in the YRDUA reached a state of relative saturation, and the influence of urbanization factors on the NPP diminished. In 2010, the coefficients of temperature and precipitation were relatively large at −0.56 and 0.36. In 2020, the regression coefficients of PD, GDPD, ULP, and CUL continued to decrease compared with 2015 and 2010, and they were still smaller than the meteorological factors, showing a relatively weak degree of influence.

Table 2 shows the $R^2$, adjusted $R^2$, AIC, and Moran's I values from the OLS and GWR models. The $R^2$ (adjusted $R^2$) values for the GWR model ranged from 0.42 to 0.53, surpassing those of the OLS model. Meanwhile, Moran's I and AIC from the GWR model surpassed those from the OLS model, suggesting that the GWR model is superior to the OLS model in examining the variables influencing the NPP. GWR is more appropriate for spatial regression analyses than OLS. Since the CUL regression coefficient is the largest, we analyzed the regional differences and evolution trend of the CUL regression coefficients and residual and explored whether CUL increasingly affects the spatial pattern of the NPP.

**Table 1.** OLS model analysis results of NPP-influencing factors from 1990 to 2020.

| Year | Variable | Coefficient | Standard Deviation | *t*/*z* Value | *p*-Value (>|t|) |
|---|---|---|---|---|---|
| 1990 | (Intercept) | 0.23 | 0.03 | 19.79 | 0.00 ** |
|  | PD | 0.02 | 0.17 | 0.09 | 0.93 * |
|  | GDPD | −0.55 | 0.14 | −3.93 | 0.00 |
|  | ULP | 0.35 | 0.12 | 2.83 | 0.01 ** |
|  | CUL | −0.48 | 0.34 | −3.16 | 0.01 *** |
|  | TEM | −0.37 | 0.04 | −10.23 | 0.00 ** |
|  | PRE | 0.36 | 0.02 | 22.05 | 0.05 ** |
|  | DEM | 0.35 | 0.06 | 14.20 | 0.00 ** |
| 1995 | (Intercept) | 0.27 | 0.03 | 19.08 | 0.00 *** |
|  | PD | 0.27 | 0.12 | 2.26 | 0.02 * |
|  | GDPD | −0.05 | 0.17 | −6.03 | 0.00 |
|  | ULP | 0.10 | 0.10 | 1.10 | 0.27 |
|  | CUL | −0.61 | 0.21 | −2.82 | 0.01 *** |
|  | TEM | −0.37 | 0.04 | −9.78 | 0.00 *** |
|  | PRE | 0.32 | 0.02 | 18.37 | 0.05 *** |
|  | DEM | 0.25 | 0.03 | 9.42 | 0.00 *** |
| 2000 | (Intercept) | 0.24 | 0.03 | 21.00 | 0.00 *** |
|  | PD | 0.22 | 0.13 | 1.73 | 0.08 * |
|  | GDPD | −0.48 | 0.15 | −3.22 | 0.00 |
|  | ULP | 0.12 | 0.09 | 0.02 | 0.98 |
|  | CUL | −0.53 | 0.22 | −2.47 | 0.01 *** |
|  | TEM | −0.43 | 0.04 | −12.08 | 0.00 *** |
|  | PRE | 0.33 | 0.02 | 20.20 | 0.00 *** |
|  | DEM | 0.41 | 0.05 | 11.02 | 0.00 *** |
| 2005 | (Intercept) | 0.38 | 0.03 | 24.73 | 0.00 *** |
|  | PD | −0.61 | 0.24 | −2.55 | 0.01 *** |
|  | GDPD | −0.36 | 0.07 | −5.28 | 0.00 *** |
|  | ULP | 0.17 | 0.08 | 2.09 | 0.04 ** |
|  | CUL | −0.86 | 0.17 | −5.19 | 0.00 *** |
|  | TEM | −0.42 | 0.04 | −11.77 | 0.00 *** |
|  | PRE | 0.22 | 0.02 | 14.07 | 0.00 *** |
|  | DEM | 0.51 | 0.06 | 10.28 | 0.00 *** |
| 2010 | (Intercept) | 0.29 | 0.02 | 25.28 | 0.00 *** |
|  | PD | −0.15 | 0.10 | −1.48 | 0.14 * |
|  | GDPD | −0.32 | 0.17 | −2.42 | 0.02 *** |
|  | ULP | −0.20 | 0.15 | −1.33 | 0.08 * |
|  | CUL | −0.46 | 0.25 | −0.66 | 0.11 * |
|  | TEM | −0.56 | 0.03 | −13.91 | 0.00 *** |
|  | PRE | 0.36 | 0.02 | 23.02 | 0.00 *** |
|  | DEM | 0.42 | 0.01 | 16.35 | 0.00 *** |
| 2015 | (Intercept) | 0.25 | 0.02 | 25.25 | 0.00 *** |
|  | PD | −0.13 | 0.13 | −1.00 | 0.32 |
|  | GDPD | −0.13 | 0.05 | −2.71 | 0.01 *** |
|  | ULP | −0.09 | 0.05 | −1.72 | 0.08 * |
|  | CUL | −0.29 | 0.13 | −1.40 | 0.16 * |
|  | TEM | −0.41 | 0.03 | −12.83 | 0.00 *** |
|  | PRE | 0.23 | 0.02 | 10.51 | 0.00 *** |
|  | DEM | 0.49 | 0.05 | 9.24 | 0.00 *** |
| 2020 | (Intercept) | 0.31 | 0.01 | 22.13 | 0.00 *** |
|  | PD | −0.11 | 0.25 | −1.03 | 0.28 |
|  | GDPD | −0.15 | 0.08 | −2.34 | 0.01 *** |
|  | ULP | −0.16 | 0.06 | −1.65 | 0.059 * |
|  | CUL | −0.32 | 0.11 | −1.58 | 0.14 * |
|  | TEM | −0.52 | 0.04 | −11.54 | 0.01 ** |
|  | PRE | 0.31 | 0.01 | 10.36 | 0.00 *** |
|  | DEM | 0.52 | 0.06 | 8.87 | 0.00 *** |

Note: *, **, and *** indicate significance at 90%, 95%, and 99% confidence levels, respectively. Abbreviations: population density (PD); GDP density (GDPD); urban land percent (ULP); comprehensive urbanization level (CUL); temperature (TEM); precipitation (PRE); digital elevation model (DEM).

Figure 7 shows the spatial impact (regression coefficients and residual) of CUL on NPP from 1990 to 2015. In 2000, the regression coefficient of CUL was roughly centered on the western part of Hangzhou, the regression coefficient becomes smaller as it goes outward, and the influence of comprehensive urbanization level (CUL) on NPP gradually decreases. In 1995, the regression coefficient for both the high-value area and low-value area shifted slightly northward towards the traffic line connecting Shanghai and Nanjing, compared to 1990. This indicates that Hangzhou's e-commerce, Internet development, and other tertiary industries gradually influenced the surrounding areas such as Shanghai and Nanjing [52]. The spatial pattern of the regression coefficients in 2000 was basically the same as that in 2005. The areas of Taizhou and Wenling in the southeast corner changed from median

regression coefficients to high values. This is because the degree of urbanization in Taizhou and Wenling was strengthened and it had begun to have a greater impact on the spatial NPP pattern.

**Table 2.** OLS and GWR model results of NPP-influencing factors from 1990 to 2020.

| Parameter | Model | 1990 | 1995 | 2000 | 2005 | 2010 | 2015 | 2020 |
|---|---|---|---|---|---|---|---|---|
| AIC | OLS | −1794.51 | −1499.15 | −1892.90 | −1484.85 | −1220.41 | −1592.63 | −1975.63 |
| | GWR | −1880.94 | −1619.63 | −2063.51 | −1648.74 | −1311.01 | −1687.76 | −1653.41 |
| R² | OLS | 0.46 | 0.43 | 0.48 | 0.47 | 0.50 | 0.42 | 0.46 |
| | GWR | 0.30 | 0.48 | 0.55 | 0.54 | 0.55 | 0.46 | 0.43 |
| Adjusted R² | OLS | 0.45 | 0.42 | 0.47 | 0.47 | 0.50 | 0.42 | 0.44 |
| | GWR | 0.49 | 0.47 | 0.53 | 0.52 | 0.53 | 0.45 | 0.50 |
| Moran's I | OLS | −0.04 | −0.06 | −0.03 | −0.05 | −0.01 | −0.03 | −0.02 |
| | GWR | −0.16 | −0.17 | −0.15 | −0.17 | −0.18 | −0.17 | −0.14 |

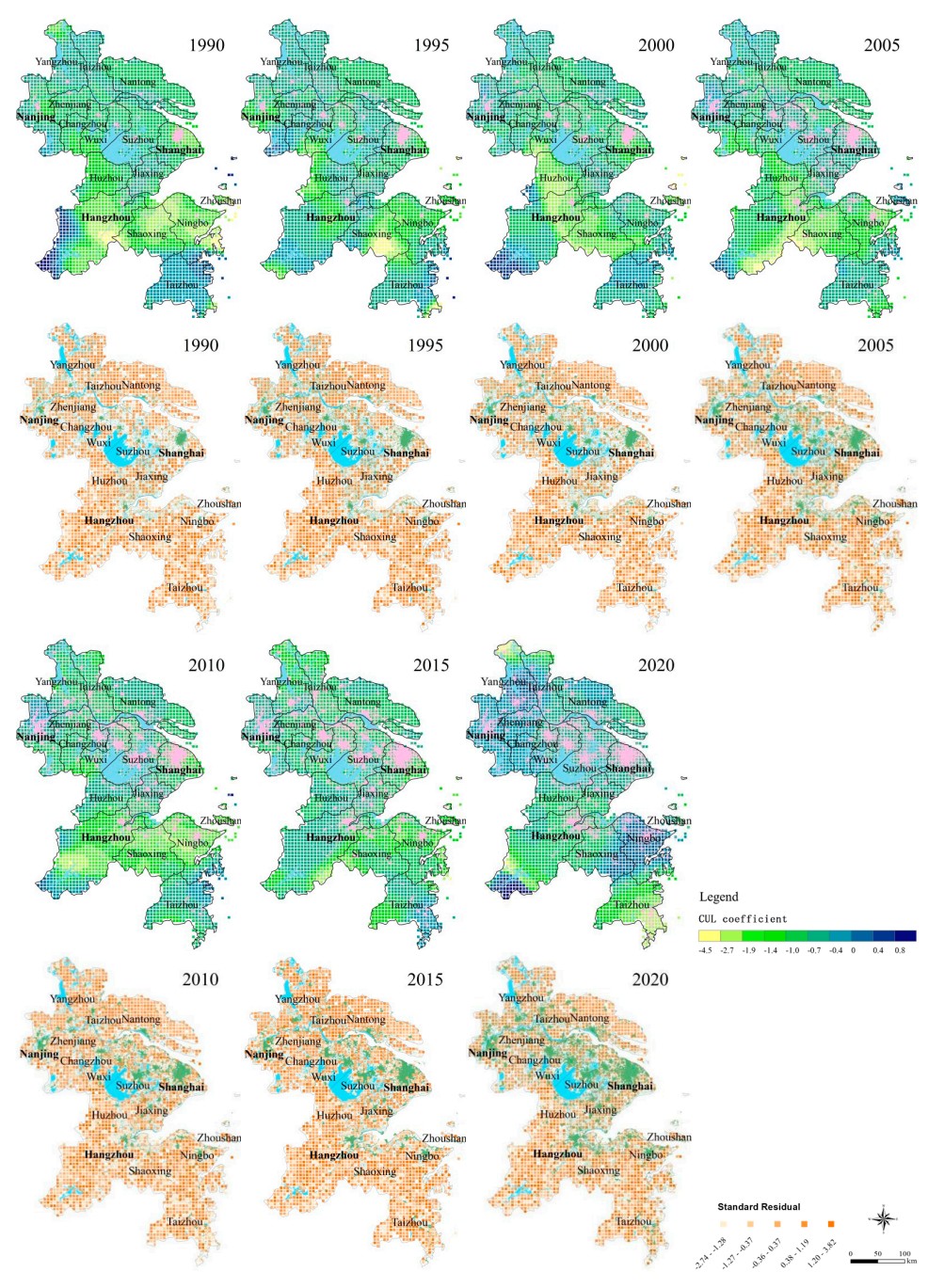

**Figure 7.** Regression coefficient and standard residual distribution of CUL from 1990 to 2020.

## 4. Discussion

### 4.1. Geographical Spillover Consequences in the Correlation between Urbanization and NPP

Geographical spillovers occur when the proximity of one unit to its nearby units influences its benefits or costs [53]. Statistically significant bivariate global Moran's I values were observed in all cases (Figure 5). This suggests that there was a significant spillover effect on the geographical correlation between urbanization and NPP in the YRDUA. Additionally, all bivariate global Moran's I values were negative, indicating that urbanization resulted in negative externalities for NPP. The findings of the bivariate LISA analysis (Figure 6) indicate that the spatial spillover process was not spatially independent. Furthermore, the transmission of the spatial spillover impact across grid cells was severely limited by the regional context [54]. In other words, when a grid is next to a highly urbanized grid, there is a higher likelihood of its NPP dropping. Conversely, when it is next to a grid with low urbanization, the reverse scenario occurs. The bivariate LISA analysis revealed distinct clustering patterns of high–high and low–low associations, indicating the need for more investigation into the relationship between NPP and urbanization. It is important to include other characteristics such as plant cover, water cover, terrain, and soil, since these may also have an impact. These variables, in conjunction with urbanization, influenced the alterations in the regional NPP [55].

The bivariate LISA diagram displays many conspicuous characteristics (Figure 6).

1. Examining the correlation between NPP and urbanization, it was evident that from 1990 to 2020, the geographical arrangement of regions deemed as not significant remained consistent. These areas are mostly located on the outskirts of metropolitan agglomerations. This is due to the fact that, compared to the Shanghai, Suzhou–Wuxi–Changzhou, and Nanjing metropolitan areas, these places exhibited a lower level of urbanization activities, such as population concentration, economic investment, and land development. Hence, urbanization is not the primary influence on NPP on the outskirts of metropolitan agglomerations;

2. In 2010, the geographical correlations of NPP and urbanization were highest in the high–high and low–high areas. The regions at the highest elevations are mostly located in the inner region, while the regions at lower elevations are found around the urban built-up areas. Prior to 2010, the rate of urban expansion exhibited a consistent and steady increase. Since 2010, there has been a growing awareness at both the national and regional levels of the rapid expansion of urban agglomerations and the environmental pollution issues associated with economic development and high-energy-consuming industries. These factors have significantly contributed to global climate change and the degradation of the ecological environment. Consequently, regulations have been implemented to regulate the unrestricted expansion of urban areas and reconfigure energy-intensive businesses in order to transition and enhance the use of clean energy sources. Hence, starting in 2015, the association between NPP and urbanization seemed to diminish;

3. The geographical distribution patterns of NPP and urbanization in 2000 and 2010 exhibited a significant degree of similarity. This outcome aligns with the findings of Qiu's study. He discovered that the decade spanning from 2000 to 2010 had the highest rate of urbanization and the greatest stability for the urban agglomeration of the Yangtze River Delta. During this time, the urban agglomeration underwent a phase of creation and development, with a consistent and continuous increase in the level of urbanization [56]. Simultaneously, the decline in NPP failed to attract attention and recognition for its effect and repercussions. The decline in NPP and the rise in anthropogenic carbon emissions are not being effectively managed and regulated. NPP and urbanization exhibited a strong geographical correlation, indicating a consistent pattern of spatial clustering.

*4.2. NPP and CUL Spatial Link Implications for Urban Agglomeration Development Programs*

Using an OLS model, our research quantitatively analyzed the influence mechanism of the spatial–temporal evolution of NPP in the YRDUA. The findings indicate that the impact of different variables on the spatial–temporal development of NPP varied across different time periods, and the influence of CUL on NPP showed an inverted "U" pattern. Coordinating regional development, adapting measures to local conditions, maximizing the benefits of regional development, and achieving a balanced, coordinated, and sustainable urban agglomeration regional development model are important goals for the progress of the YRDUA region. Prioritizing the low-NPP areas that are most vulnerable to urbanization is crucial when embarking on urban development projects. Any region exhibiting a substantial NPP should be designated as an ecological reserve to prevent or limit urban expansion [57,58].

The regression analysis of the CUL in 2015 revealed that the regions with high regression coefficients were concentrated in the western border region of the urban agglomeration, as seen from the distribution patterns of Shanghai and Hangzhou. This is mostly due to the urbanization growth of the urban agglomeration, which has started expanding towards the west. As a result, it has begun to link with and influence the urbanization process of the western strip region of the urban agglomeration [59]. As far as the actual situation is concerned, in the "Yangtze River Delta Urban Agglomeration Development Plan (2015–2030)", 10 prefecture-level cities under the jurisdiction of Anhui Province, which is close to Jiangsu, have also been assigned to the YRDUA, so that the YRDUA has a more solid development foundation and geographical space, which can better realize the improvement of quality and efficiency and the integrated development of large regions [60].

The study findings can more accurately align with the current state of development in the YRDUA. Additionally, the research conclusions may serve as a foundation for the creation of regional development policies for the YRDUA. The analytical framework not only emphasizes the application of new methods but also pays attention to the dynamic space of regional cooperation and its interconnections [61]. It aims to provide support for further coordinating regional balanced development and strengthening regional exchanges and cooperation by identifying and quantifying the spillover effect between regions. This research approach can also provide new research ideas and methods for other domestic regional economic development research [62].

*4.3. Limitations of the Applied Method*

However, there are still some limitations in this study. One problem is that the $R^2$ values are relatively low for both the OLS and GWR models, suggesting that other factors affecting NPP were not fully explored. Further research should develop improved models based on the characteristics of the study area to analyze the spatial–temporal evolution of NPP, such as spatiotemporal weighted regression (STWR) models. The STWR model is characterized by using a novel "time distance" for weighting to capture fine spatiotemporal heterogeneity, as opposed to the traditional geographically and temporally weighted regression (GTWR) approach [63]. These advanced spatiotemporal modeling techniques could help to better elucidate the relationship between NPP and its driving factors, including the role of past carbon storage data to determine the relationship between social behavior and natural resources. Another issue is that the geospatial data layer should be further refined in future research. It is worth noting that the assessment of ecosystem services depends on the choice of proxy indicators. In our study, the NPP indicator, which characterizes carbon fixation and oxygen release, was selected to reflect the regulatory function of ecosystem services. If the NPP data based on MODIS data cannot be accurately spatially predicted, or the collected data do not have sufficiently high spatial and temporal resolutions, there may be errors in the results of the correlation analyses and regression fittings. This issue also arises in proxies used for evaluating urbanization, and all three levels have inherent issues concerning the quality and accuracy that may impact the study's findings.

The findings indicate that GWR outperforms the standard regression method (OLS) in explaining the relationship between urbanization and the responsiveness of ecosystem services, especially for NPP. The results derived from the methodologies used in this investigation are very reliable and satisfactory, indicating that spatially explicit modeling approaches could be valuable for decision making and policy formulation. If the data used in this investigation can be gathered for other geographical areas, the technique could be applied in those areas to evaluate indicators of urbanization and NPP and perform a geographical examination of the interconnections between these two aspects. The purpose of this research is to assist governments in making informed choices that will contribute to the long-term sustainability of urban agglomeration areas, including the economic, environmental, and sociocultural aspects.

## 5. Conclusions

Our research investigated the relationship between ecosystem services (with NPP representing regulatory functions) and urbanization from a geographical standpoint, taking into account the spatial correlations and dependencies. The findings of our investigation led to the following conclusions: (1) The bivariate global Moran's I of urbanization and NPP from 1990 to 2020 exhibited negative values, suggesting a global negative connection between the two variables. From a local standpoint, there was a geographic disparity in the correlations between CUL and NPP. The bivariate LISA approach identified and presented four different types of local correlations (namely, high–high, high–low, low–high, and low–low) between NPP and urbanization. (2) The spatial regression analysis revealed that urbanization and other influencing variables have varying effects on NPP. Due to the growing urbanization, NPP reached its peak in 2005, with 1995 and 2010 following closely after. The correlation was lowest in 2015. (3) Aside from urbanization, environmental services are also influenced by other variables such as climate and geography. When accounting for spillover effects in the regression analysis, the influence of urbanization on ecosystem services showed a steady increase from 1990, followed by a gradual decline after 2010, in contrast to the findings obtained using the OLS method. However, our study has certain limitations. First, we used NPP as a proxy for ecosystem services, which may not fully capture all aspects of regulatory functions. Second, the spatial regression models employed may not account for all potential confounding factors and complex interactions. Third, our analysis focused on the national scale, and finer-scale local variations may exist. Despite these limitations, our findings offer practical implications for urban planning and industrial site selection, highlighting the importance of considering spatial dependencies and trade-offs between urbanization and ecosystem services. Future research could explore alternative measures of ecosystem services, incorporate additional explanatory variables, and conduct multi-scale analyses to further refine our understanding of this complex relationship. We recommend that urban planners and policymakers consider the spatial patterns and correlations identified in our study when developing urbanization strategies and industrial zoning plans. Balancing economic growth with ecosystem conservation requires a holistic approach that accounts for geographic variations and spatial spillover effects. Integrating these considerations into decision-making processes can promote sustainable urban development and environmental stewardship.

**Supplementary Materials:** The following supporting information can be downloaded at: https://www.mdpi.com/article/10.3390/land13040562/s1, Figure S1: (a) Spatial pattern of population (POP) in the YRDUA from 1990 to 2020. (b) Spatial pattern of gross domestic product (GDP) in the YRDUA from 1990 to 2020. (c) Spatial pattern of urban land percentage (ULP) in the YRDUA from 1990 to 2020.

**Author Contributions:** Conceptualization, methodology, writing—review and editing, J.G. and M.L.; formal analysis, writing—original draft preparation, visualization, J.G.; supervision, M.L. and X.W. All authors have read and agreed to the published version of the manuscript.

**Funding:** This work was supported by the National Natural Science Foundation of China (42371219), Gansu Province Philosophy and Social Science Planning Project (2023QN010), and the Young Teachers' Research Ability Enhancement Program Project (NUNW-SKQN2023-24).

**Data Availability Statement:** Data are contained within the article.

**Acknowledgments:** The authors are grateful to the editor and reviewers for their valuable comments and suggestions.

**Conflicts of Interest:** The authors declare no conflicts of interest.

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
