# Peer review of "Unveiling the Impact of Urbanization on Net Primary Productivity: Insights from the Yangtze River Delta Urban Agglomeration"

_land, doi:10.3390/land13040562_

Round 1
Reviewer 1 Report
Comments and Suggestions for Authors
Interesting paper on how to evaluate the spatio-temporal effect of urbanization on the carbon cycle interesting in its discussions and for further work, with the example of a Chinese region subject to urbanization, that can be applied to other regions of the world.
I suggest accepting publication after minor revision (corrections to minor methodological errors and text and english editing). My comments and doubts are:
p3
L139 0.68 billion tons of ? units? specify units.
p4
L142 143 parenthesis missing
L142 - 145 Figure 1 caption: left and right are inverted, space before parenthesis
p5
L187-188 “Due of the strong resemblance in the geographical pa ern of PD, GDPD, and ULP, these three variables were combined into a single indicator known as comprehensive urbanization level (CUL). ”
What do you mean by “resemblance”? “correlation”? If so, why use the 3 index if they are strongly correlated (do you know it before, or is it a result …seen in section 3.3.?). Why average them? Specify or explain.
Equation 1 doesn't mention time. Question: "𝑈𝑖,𝑚𝑖𝑛 represent the highest and lowest values, respectively, of the i-th urbanization 194 indicator over all grids."... and over all years? Specify.
p6
L230 "The text continues here." ? check redaction
L231 “NPP effect on”..... would “ NPP effect related to” be better?
p7
L243 “Our study using Guass kernel “ replace by “Our study uses Guass kernel “
L265 check last sentence redaction or verb tense.
L252 space after Fig.
Figure 3 has a wrong legend.
In section “3.1. Spatial Patterns of CUL in YRDUA “ :
From section "2.3.1. Urbanization Assessment " it is said that indexes were subjected to range standardization in order to convert their values into a uniform range of 0 to 1. These standardized values were then averaged to get the CUL value ... “
I understand that the average of 3 values ranging from 0 to 1 would also be values between 0 and 1. But here in paragraph text of 3.1 CUL values are higher than 1 (L256 the CUL increased from 67.37 to 105.75 )... but in Figure 4 reaches CUL values 1200. What are the units? Explain or specify…
In FIgure 4, Zhoushan has a very small CUL value in 2000, so rate "1995 2000 year" should be negative. but it is 10%in figure 4 use round numbers, 10, 20
In FIgure 4, will it be clearer if we delete negative CUL values in the superior axis?
p10 L297 seven-year “sample” not “period”.
p11
L320 delete comma after "showed"
L322 Confuse. Rephrase "Except for the regression coefficients of PD in 2000 and the ULP in 2005 were positive correlation, , the other factors from 1990 to 2020 were all negatively correlated with NPP" by "Except for the regression coefficients of PD in 2000 and the ULP in 2005 that showed positive correlation, all factors from 1990 to 2020 were all negatively correlated with NPP" .
I suggest listing "the other factors" between parentheses.
L325: A little confusing. "The coefficients for each year" … which year are you mentioning? in the text, sentence only 4 values are mentioned...?
L326 -1.08 should be -0.48 ?
L330 Replace "At 2010, The" by "In 2010, the" , preposition and capital letter .
Table 1:
Adding lines between years could help read Table 1.
1990 is missing in table 1. (entry 1, entry 2 ?).
Replace "Title2Variable" and "Title3Coefficient".
L333 Replace "shows its degree of in uence is relatively weak." by "showing a relatively weak degree of influence."
L336 No need to put 1 in " 1 Note: "
p13
L337-344 Font size of the whole paragraph is of caption not of text.
L341-344 Rephrase sentence.
L346 homogenize the use of space after Fig in document: “Fig.7” “Fig. 7”.
Comments on the Quality of English LanguageSee comments.
Reviewer 2 Report
Comments and Suggestions for Authors
The manuscript presents a study that analyzes the spatiotemporal relationship between NPP and urbanization. Although the data and the methods used are impressive, the findings are not clearly presented in the work. Despite the impressive data, the usage of those data in the models are not clearly introduced. How were the tools used in the study? How to interpret them? These are important and should be clearly discussed. It almost seems like the authors fed data into a toolbox and obtained the results without giving time to clearly present their interpretations. There are also multiple sections that simply praise the paper and has the meaning except maybe to elongate the discussion (?). Because of the lack of evidence or clear new knowledge presented in this work, this work requires major revision or resubmission. Kindly read through the following comments:
1. Clear definition of Net Primary Productivity (NPP). It is mentioned in L54 without any description. What does this mean? Why is it necessary to investigate it? How is it calculated and what are the limitations/uncertainties of their estimates?
2. Abstract issues:
a. L22: "at the grid size" suddenly appears. What does this mean? There is no discussion on grid resolution dependencies throughout.
b. L22: "investigates the temporal and geographical features and related patterns of NPP and their effect mechanisms" is unclear. What are these "effect mechanisms"? How is it defined?
c. L23: "The findings of this study are informative and have practical applications." is an unnecessary statement in the middle of the abstract.
d. L26: What are these "data sources"?
e. L30: "urbanization on a global scale" this is a critical error for an abstract (also mentioned in L285). Many readers of papers only read abstracts but this work does not focus on global scale analyses at all.
f. L34: "geographical relationship" there is little discussion within the paper about geography. What do the authors mean by this?
3. L63 needs a citation. Specifically on "Ecosystem services are categorized into four distinct groups". What are these?
4. L78: The statement is confusing since the authors of the paper did not really use these approaches, "The models used in this study consistes of statistical models, process-based models..."
5. L102: The research gap includes "clustering relationship" but what does this mean?
6. One of the objectives (L121) is about using bivariate global and local Moran's I but there is no clear relation between this and the contents of the Introduction.
7. The datasets should be more specific and their purpose:
a. Which MOD17A3 was used? Proper citation of the data is needed.
b. L157, what was the approach of estimating the LULC? this is crucial.
c. L164, interpolation process is unclear. Which resolution?
d. L166, specific name and source of the DEM used. (this dataset usage is not very clear in the discussion)
e. GDP, population density, and climate were collected to estimate CUC but specific information is needed. Who were the providers and how were they obtained?
8. Sect. 2.3.2, Sect. 2.3.3, talks about the statistical approaches in general. They should be more specific. How were the data fed into these models? How were the parameters obtained? How should its outputs be interpreted? What about their limitations?
9. L210. Source of "GeoDa 1.12" what is it?
10. What are the units for 67.37 and 105.75 in L256? What is its physical meaning?
11. L266 onwards. What is the meaning of "openness of urban agglomeration"? What are "vertical connection mode" and "horizontal connection mode"? How can that be inferred from the model outputs? L269 to L271 is interesting but there is no citation that adds evidence to this statement.
12. The figures are unclear and difficult to interpret. The discussion that is supposed to be supported by these figures can be difficult to interpret.
13. "LISA" is suddenly included in the analyses. How should this be interpreted? What is the meaning of "high-high", "low-low", and so on? This has to be described or the findings in L383 to L416 won't be convincing.
14. Sect. 3.3 shows the usage of OLS. It is unclear how they are calculated. This analyses do not discuss about the cluster but for each individual grid. The discussion also shows the results but there is little interpretation as to why the results are so. Interpretations backed by other relevant works might be needed.
15. Table 1 is unclear. What is "Title1, Title2, entry 1, entry 2"?
16. L419 to L421 has no bases. How can we interpret this "U" type from the figures? Where is this findings coming from.
17. L429 to L439, L452 to L458, L483 to L493 simply evaluates the model approaches used in this study. This not directly related to the objectives. Consider removing this part or shifting this elsewhere.
18. In the conclusions, L504 "The findings of our investigation corrobate the conclusions drawn from subsequent empirical analysis" is an unclear statement? What is the subsequent empirical analysis?
19. L510 to L512 statement talks about why OLS is better than GWR but this does not directly link with the objectives or research gaps.
20. L516, "Our analysis elucidates the ramifications of the interplay between environmental services and urbanization." is also unnecessary and simply complimenting this paper. Conclusions should directly focus on the scientific findings and meeting the objectives.
Comments on the Quality of English LanguageKindly read my comments above.
Reviewer 3 Report
Comments and Suggestions for Authors
This study explores the spatiotemporal relationships between the urbanization and NPP and from 1990 to 2020. I have the following concerns: (1)The introductory section is suggested to be rewritten in a clearer and more concise form. The current manuscript is lack of literature reviews on the relationship between urbanization and NPP(including its calculations) (2) What is the resolution of LULC? Are multiple sources of data at different resolutions used? How to make the resolution consistent? (3) Lines 177-188: The calculation for banization level (CUL) should be given. (4)There is no data to support this statement "Shanghai and Hangzhou have maintained a stable growth rate due to the development of emerging high-tech industries." (5)What is the ''best' of Moran’s I ? The changes of Moran’s I can be included. (6) The time should be added to the Fig.5 (7) The readability of Table 1 is very poor, and it is hard to tell the time changes of the values. (8)Most of the R2 in Table 2 are lower than 0.5, which may indicate that the model may be invalid or fail the statistical tests.(9) No relevant data is shown for "e-commerce, Internet development, and other tertiary industries" to confirm "This indicates that Hangzhou's e-commerce, Internet development, and other tertiary industries have gradually influenced the surrounding areas such as Shanghai and Nanjing." (10) Spatial variation in residuals is ignored. Changes in residuals may also be important for this study. (11)What does the CUL of Figure 7 mean? (12) Where does it show that "the influence of CUL on NPP is inverted "U" type"? (13)Taking administrative districts as spatial subjects may limit the practical feasibility of incorporating NPP impacts into comprehensive regional landscape design and industrial layout of urban agglomerations. However, although Line 384 was mentioned in the discussion, there was no comparative experiment or paper support. Perhaps using administrative districts is more in line with actual planning needs?(14)The indicators used in calculating the urbanization index in this study are not comprehensive enough, and there are few natural factors selected; the only models compared in the experiment are OLS and GWR, and there is no need to improve the model based on the characteristics of the study area. A discussion on the possibilities of STWR(Spatiotemporal weighted regression) applications was suggested.
Comments on the Quality of English LanguageModerate editing of English language required
Round 2
Reviewer 2 Report
Comments and Suggestions for Authors
The authors have kindly addressed my comments, and it has been significantly improved. Upon inspection of the revised manuscript, the authors should clarify further on the following:
1. The sentences that follow after "The purpose of this study..." in L198 may be a new paragraph. It helps the readers identify the focus.
2. The data sources must be clearly cited. For example, it is not appropriate to simply state the GDP density and population density can be downloaded in www.resdc.cn. This dataset is important for the reliability of the outputs and it is not common to represent GDP at 1-km resolution. The authors should give more specific locations and descriptions for this. The same goes for the air temperature and precipitation. Interpolation at 1-km is the final step. The authors should mention the details of these weather stations (how many they are and how spread apart they are). There are other datasets of similar treatments.
3. Eq. 2 is an assumption in the study to generate the CUL. It will help to discuss the bases for this and to include figures similar to Fig. 3 for PD, GDP, and ULP.
4. L 1018 "significant spatial correlation" cannot be seen from Fig. 5. This needs to be rephrased.
5. It is strange that the section on the conclusions is shorter than the abstract itself. It is necessary to summarize the limitations / assumptions / recommendations as well.
Comments on the Quality of English LanguageKindly reread the manuscript again and keep improving on clarity. For example, there is no reference for the word "Anselin" in L760. What is it?
Reviewer 3 Report
Comments and Suggestions for Authors
This manuscript has significantly improved after revision. However, some minor revisions are needed.
1: Reference order is not alphabetical.
2:I mentioned the Spatiotemporal weighted regression (STWR) rather than geographically and temporally weighted regression (GTWR)( "geographical temporal weighted regression" is the wrong spelling). Please add discussions on STWR ("A spatiotemporal weighted regression model (STWR v1.0) for analyzing local nonstationarity in space and time"). It is characterized by using a novel "time distance" for weighting to capture fine spatiotemporal heterogeneity.
3: It is recommended that you check your constant terms(Intercept)in all models to ensure that they do not dominate(not significantly larger than other coefficients) the fitted model. Because they can indicate whether the interpretability of the coefficients is still sufficient, making the analysis more robust.
Comments on the Quality of English Language
Minor editing of English language required.
